# Synthesis of a Metal Oxide by Forming Solvate Eutectic Mixtures and Study of Their Synthetic Performance under Hyper- and Hypo-Eutectic Conditions

**Omar Gómez Rojas [1]**, **Simon R. Hall [2],\*** **and Tadachika Nakayama [1],\***

[1]  Nagaoka University of Technology, 1603-1, Kamitomioka Nagaoka, Niigata Prefecture 940-2188, Japan; omargr@vos.nagaokaut.ac.jp

[2]  Complex Functional Materials Group, School of Chemistry, University of Bristol, Bristol BS8 1TS, UK

\*  Correspondence: simon.hall@bristol.ac.uk (S.R.H.); nky15@vos.nagaokaut.ac.jp (T.N.)

**Abstract:** The synthesis of $YBa_2Cu_3O_{7-x}$ (YBCO or 123) superconductor was carried out under hyper- and hypo-eutectic conditions with different ammonium compounds, i.e., ammonium nitrate, formate, acetate, carbonate, bicarbonate, and tetramethylammonium nitrate. The aim was to find more affordable synthetic pathways using highly available and cheaper compounds, as well as to study the crystal formation under no-carbon conditions when ammonium nitrate was employed. Best results were obtained when eutectic conditions were achieved, namely by ammonium nitrate and YBaCu nitrates in a 5:1 molar ratio (81% of the superconductor). Ammonium formate, acetate, carbonate, and bicarbonate did not produce eutectic mixes. Temperature analysis of the reaction carried out by ammonium nitrate/YBaCu nitrates indicated the formation of barium carbonate, despite no carbon source being used in this reaction. This phenomenon is further discussed in this work. Consequently, tetramethylammonium nitrate, as a chelator and carbon source, was used, providing >96% of the superconductor.

**Keywords:** metal-containing eutectic mixtures; high-temperature superconductor; metal oxide; ammonium compounds

## 1. Introduction

The synthesis of metal oxides has been an intensive research field due to the plethora of properties that these materials can provide [1]. For this reason, synthetic procedures have been constantly evolving to create new, faster, less time- and energy-consuming routes. Different approaches have been found to be successful such as wet routes, either surfactant-free [2] or assisted by a liquid like ethylene glycol [3], hydrothermal [4], ionothermal [5], sol–gel [6], and microwave-assisted [7] syntheses, just to mention a few. Every synthetic procedure has advantages and disadvantages, but in order to reduce the latter, more complex approaches have been taken, for example, the combination of sol–gel and hydrothermal methods which have achieved interesting results [8,9], or the use of ionic liquids as solvents and chelators and an organic source as the coagulant agent as well as non-specific chelating compound [10–13]. Despite the success that the ionic liquids have shown in different areas via ionogels [14], as electrolytes [15], catalysts [16], or as solvents for metal extraction [17], the use of such for the synthesis of complex metal oxides has not seen great diversification, partly due to the uncertainty around their role in the synthesis. A recent study, however, has provided insight into their behavior at early stages of the synthetic protocol [12]. Another matter to be solved is the elevated price that ionic liquids tend to have, for example, 1-Ethyl-3-methylimidazolium acetate (≥95.0%) sold by Sigma

Aldrich costs the equivalent of ¥54,500/100 g. As a result of their expense, variations of these solvents have been developed, named deep eutectic solvents. With similar behavior and chemical interactions, these solvents provide cheaper and easier to synthesize alternatives and have already found success in the synthesis of metal oxides [18,19]. In this work, instead of solubilizing metal compounds in a deep eutectic solvent, we used the metal compounds themselves as eutectic forming agents, significantly reducing the expenses related to reactants (tetramethylammonium nitrate ¥29,600/100 g, tetramethylammonium formate ¥37,357/100 g, ammonium nitrate ¥1664/100 g, ammonium acetate ¥1828/100 g, ammonium carbonate ¥6120/100 g, ammonium bicarbonate ¥3640/100 g, and ammonium formate ¥12,200/100 g; all the prices shown are from Sigma Aldrich). This is analogous to the solvation of metal compounds in ionic liquids [20]. Here, the synthesis of the high-temperature superconductor $YBa_2Cu_3O_{7-x}$ (YBCO) oxide is investigated with a variety of different compounds such as ammonium nitrate, ammonium formate, ammonium acetate, ammonium carbonate, ammonium bicarbonate, and tetramethylammonium nitrate. Furthermore, the synthesis is also investigated under hypo- and hyper-eutectic conditions in order to examine the effect of this variation in the pursuit of high yields of YBCO.

## 2. Materials and Methods

### 2.1. Materials

Tetramethylammonium nitrate (425257), tetramethylammonium formate (438375), ammonium nitrate (A9642), ammonium acetate (A1542), ammonium carbonate (207861), ammonium bicarbonate (09830), ammonium formate (516961), barium nitrate (217581), and copper(II) nitrate trihydrate (V800130) were purchased from Sigma-Aldrich Japan, and yttrium(III) nitrate hydrous (150316) was purchased from High Purity Chemicals Japan. Barium nitrate (021-09035, 99.9%) was purchased from Wako Japan. Deionized water was obtained using a MilliQ PureLab Ultra (18.2 M $\Omega$ cm$^{-1}$). None of the materials required further purification and were used as received.

### 2.2. Generation of Metal-Containing Deep Eutectic Solvents

For the generation of metal-containing deep eutectic solvents, every ammonium compound, namely ammonium nitrate, formate, acetate, carbonate, bicarbonate, and tetramethylammonium nitrate, and formate was mixed with the metal nitrates, viz. yttrium nitrate, barium nitrate, and copper nitrate, in a range of 20:1 to 1:20 ammonium compound/metal nitrate. Initially, the aqueous solutions were mixed correspondingly, and left at 70 °C for a period of 24 h to ensure full dehydration. After that, solid products were obtained in all the mixes of ammonium compounds with every single metal nitrate, in all the range of molar ratios. The eutectic point was then noted by raising the temperature, using a Fisher Scientific Isotemp Advanced stirring hotplate, until the solid became entirely liquid. However, from all the ammonium compounds used, eutectic mixtures could only be produced using ammonium nitrate and tetramethyl ammonium nitrate in interaction with yttrium nitrate, copper nitrate, and barium nitrate. When ammonium nitrate was employed, copper nitrate and yttrium nitrate gave eutectic mixtures at the same molar ratio, i.e., 5:1 ammonium nitrate/copper–yttrium nitrate. However, barium nitrate exhibited eutectic behavior on the molar ratio of 20:1 ammonium nitrate/barium nitrate. The use of tetramethyl ammonium nitrate provided the same results as observed for ammonium nitrate.

### 2.3. Generation of Aqueous Precursors

Yttrium nitrate (0.05 M), barium nitrate (0.1 M), and copper nitrate (0.15 M) were individually mixed in vials with deionized water under stirring until all salts were dissolved. Similarly, ammonium compounds, i.e., ammonium nitrate (0.5 M), formate, acetate (0.5 M), carbonate (0.5 M), bicarbonate (0.5 M), and tetramethylammonium nitrate (0.5 M), and formate (0.5 M), were also dissolved under constant stirring with DI water until all the solid material was fully dissolved.

*2.4. Heating Protocols*

All the mixes of ammonium compounds with metal nitrates were calcined following the same calcination route: max temperature 920 °C, at a ramp rate of 5 °C/min with a dwell time of 2 h.

*2.5. Characterization*

For the characterization of the samples FE-SEM JEOL JSM 67000F, powder X-ray diffraction (pXRD) was carried out on a Rigaku RINT2000 diffractometer (CuKα 1 radiation at λ = 1.5418 Å). Rietvield analysis was done via Profex 3.12.1 Software [21]. Diffraction patterns were analyzed using the Inorganic Crystal Structure Database (ICSD) reference numbers for phase identification (Table S3). Size distribution was done via ImageJ software [22]. Thermogravimetric analysis was performed on a Bruker TG-DTA2000SA.

## 3. Results and Discussion

*3.1. Eutectic Mixtures*

### 3.1.1. Ammonium Nitrate

With the aim of reducing cost expenses of chemical compounds and to analyze the crystal growth in the absence of carbon sources, ammonium nitrate was chosen. Eutectic mixtures were formed between metal nitrates and ammonium nitrate. Individual metal nitrates were combined with the molecule, and their eutectic points noted, 105 and 118 °C for copper nitrate and yttrium nitrate, respectively, both in a molar ratio of 5:1 ammonium nitrate/copper-yttrium nitrates (Figure 1). Eutectic mixes were formed with barium nitrate in a molar ratio of 20:1 ammonium nitrate/barium nitrate. Combination of YBaCu nitrate mixture and ammonium nitrate at 5:1 molar ratio resulted in a deep melting point at 100 °C, which was five degrees lower than the copper nitrate and ammonium nitrate eutectic mixture. Despite barium nitrate not forming a eutectic solvent at that molar ratio, no precipitates were observed (Figure 1).

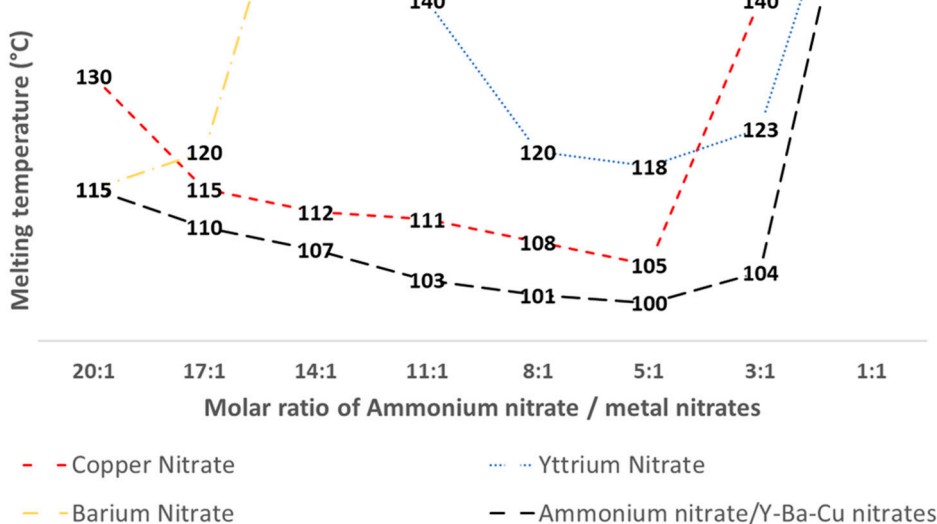

**Figure 1.** Phase diagram showing the eutectic behavior of the metal nitrates when mixed with ammonium nitrate in different molar ratios.

### 3.1.2. Ammonium Carbonate, Ammonium Bicarbonate, Ammonium Acetate, and Ammonium Formate

Ammonium carbonate, ammonium bicarbonate, ammonium acetate, and ammonium formate were also employed; however, eutectic mixtures could not be formed. In those cases, depending on

the molar ratio between the metal cation and the ammonium compound, two different interactions were observed. When the molar ratio of ammonium nitrate was higher than the metal nitrates, the ammonium was able to coordinate the metal. On the other hand, when molar ratios of ammonium compounds were lower in comparison with the metal nitrates, the carboxylic group, i.e., acetate, carbonate, bicarbonate, and formate, coordinated the metal cations. This was observed by changes in color, from dark blue to dark green. The appearance of color based on cation coordination environment agrees with previous studies [23,24]. A table summarizing the data can be found in Figure 2.

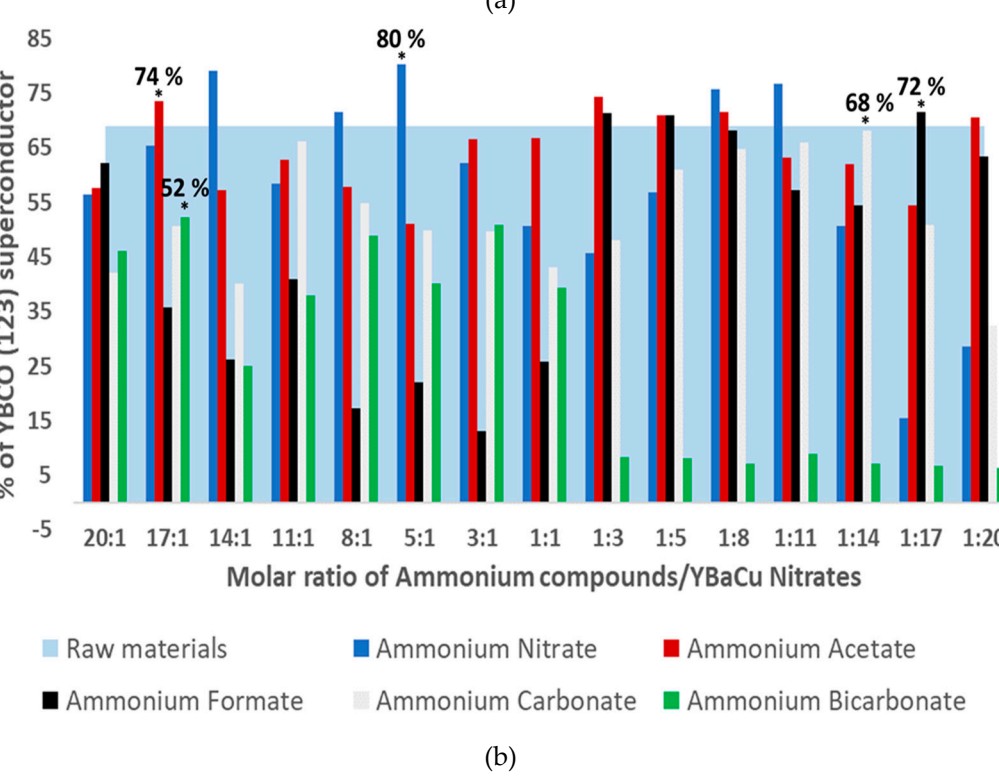

|  | Ammonium Nitrate | Ammonium Acetate | Ammonium Formate | Ammonium Carbonate | Ammonium Bicarbonate | Raw materials |
|---|---|---|---|---|---|---|
| Eutectic behaviour | Yes | No | No | No | No | N/A |
| Molar ratio | 5:1 | N/A | N/A | N/A | N/A | N/A |
| Highest superconductive phase percentage | 80% | 74% | 72% | 68% | 52% | 69% |
| Molar ratio | 5:1 | 1:3 | 1:17 | 1:14 | 17:1 | N/A |

(a)

(b)

**Figure 2.** (**a**) Tabulated data summarizing the eutectic behavior of each mix of ammonium compounds and metal nitrates (white), and maximum percentage of the superconductive phase and the molar ratio at which it was obtained (gray); (**b**) summary of the maximum percentage of the superconductive phase obtained via different ammonium-containing compounds in different molar ratios with the metal nitrates.

### 3.2. Synthesis of YBCO Superconductor

The synthesis of YBCO (123) superconductor was attempted with the mixes of ammonium nitrate, ammonium carbonate, ammonium bicarbonate, ammonium acetate, and ammonium formate

in combination with YBaCu nitrates, as well as in different molar ratios. The results can be seen in Figure 2. In none of the cases was the phase percentage of the superconductive crystalline phase above 80%, based on Rietveld refinement; for comparison, a control reaction with only YBaCu nitrates was performed, generating 69% of the total mass of the superconductive phase (Figure 2a,b). Tabulated data of every molar ratio can be found in Table S1. In regard to the syntheses with the mixes of ammonium nitrate, ammonium carbonate, ammonium bicarbonate, ammonium acetate, and ammonium formate with YBaCu nitrates, the highest percentage of the superconductor was obtained with ammonium nitrate and in the respective eutectic point previously found, namely 80% at a 5:1 molar ratio of ammonium nitrate/YBaCu nitrates (Figure 2a,b). This highlights the importance of having a highly mobile system which allows fast mass transport, and better metal to metal interactions.

Regardless of this, the target superconductive phase is still a relatively low amount. Complete tabulated data can be found in Table S1.

### 3.3. Temperature Analysis

#### 3.3.1. Via Rietveld Refinement of 5:1 Ammonium Nitrate/YBaCu Nitrates

Previous studies have shown that during the calcination, in every case, metal carbonates were formed and those crystal compositions were identified as promoters of high yields of single crystal phases [25]. The metal carbonates are formed via the interaction of the metal cation with a carboxylic coordinating sphere [12]; therefore, with the use of compounds such as ammonium nitrate, YBaCu nitrates, and no carbon source added to the reaction, the formation of such crystals should not be expected. The thermal route is shown in Figure 3. At the beginning, ammonium nitrate is almost solely observed at 120 °C, present as 88% of the material. However, due to the lower boiling point, i.e., 180 °C [26], it will immediately evaporate, and at 220 °C 83% barium nitrate and copper oxide are observed. Barium nitrate will gradually decompose from 220 to 620 °C, and copper oxide sees a gradual gain in total phase percentage from 220 to 720 °C composing 44% of the reactive material at the last temperature. However, contrary to every expectation, barium carbonate is also observed during the reaction process, highlighted with an asterisk (*) in Figure 3, first seen at 420 °C with only 8% but with a positive tendency on the next 200 °C, covering 55% and 59% of the total mass at 520 and 620 °C, respectively. It is important to remember that no sources of carbon were added to the reaction.

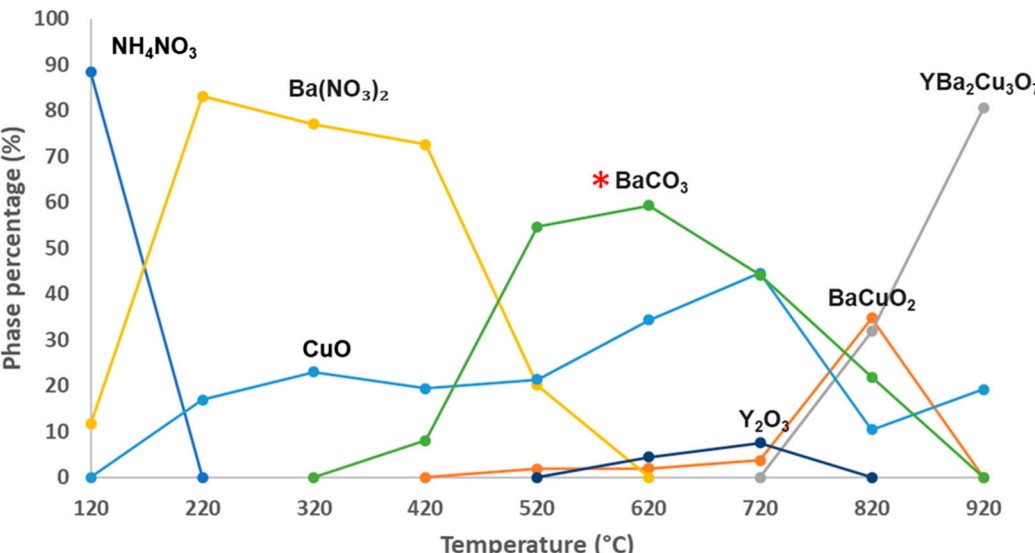

**Figure 3.** Graphical representation via qualitative analysis of pXRD patterns from the temperature analysis of the synthesis of $YBa_2Cu_3O_{7-x}$ (YBCO) (123) superconductor in the 5:1 molar ratio of ammonium nitrate/YBaCu nitrates.

To avoid any possible formation due to impurities on the crucible, syntheses with new crucibles were carried out with no changes on the outcome whatsoever. The reactants employed were of high purity (>99%) and with the best quality standards; therefore, to uncover the reason for barium carbonate formation, several experiments were performed. First, individual reactants, namely, barium nitrate, copper nitrate, and yttrium nitrate were placed in separate furnaces and new crucibles, calcined at a maximum temperature of 620 °C for 2 h, and a ramp rate of 1 °C/min. As expected, copper nitrate and yttrium nitrate provided pure phase copper oxide and yttrium oxide (Figure 4a,b). However, results with barium nitrate provided a mix of barium carbonate and barium silicate ($B_2SiO_4$), silicon being provided by the surface of the crucible. To ensure that these results were reproducible, crucibles from a different provider were used, as well as the use of barium nitrate provided by Sigma-Aldrich Japan (217581, > 99%). The product once again exhibited a mix of barium carbonate and barium silicate (Figure 4c). The observed results are due to the highly reactive nature of barium [27]; additionally, similar behavior of barium nitrates being replaced by carbonates has also been reported [28,29]. Subsequently, the reaction protocol from 520 to 820 °C saw the barium nitrate decomposition at 620 °C, the formation of yttrium oxide with 8% of the total mass percentage at 720 °C, and the presence of barium copper oxide with 35% of the total mass percentage at 820 °C. Those crystal compositions, copper oxide, barium copper oxide, yttrium oxide, and barium carbonate, reacted to form the superconductive phase with 32% and 81% at 820 and 920 °C, respectively.

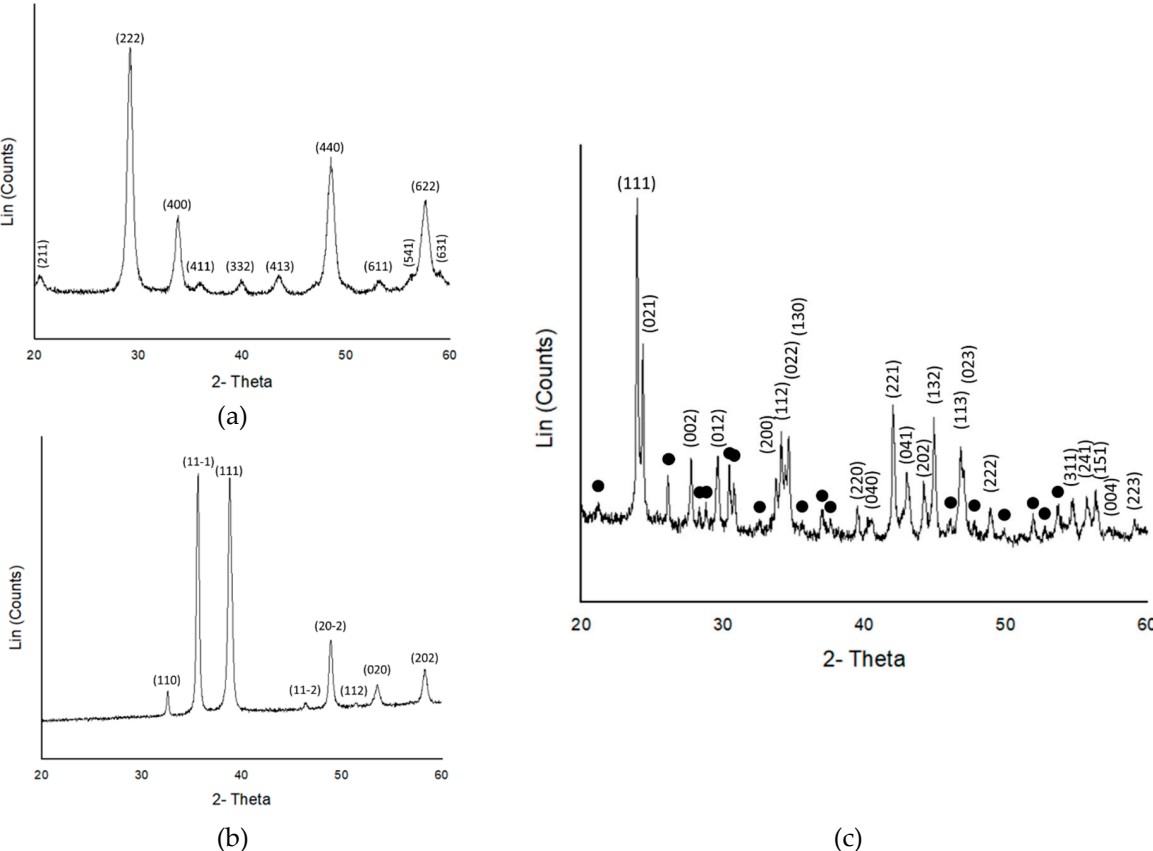

**Figure 4.** Powder X-ray diffraction patterns of (**a**) yttrium oxide, (**b**) copper oxide, and (**c**) barium carbonate and barium silicate (•). All the reactions were carried out using their respective metal nitrates, calcined in new crucibles, and a maximum temperature of 620 °C.

Based on this result, the formation of metal carbonates is essential due to the fact that those crystalline compositions thrive even under non-carbon conditions, as elements like barium can react

with carbon that the atmosphere provides. Therefore, adding chemical compounds that supply carbon to the reaction media provides faster thermal synthetic routes and higher purity [12,25].

To prove the formation of carbonates due to the interaction of barium with the carbon contained in the atmosphere, a reaction under argon atmosphere was done. The outcome showed no carbonates in the least, although only weak noise/signal ratio diffractions were observed, namely, yttrium barium copper oxide with the chemical composition $Y_{0.5}Ba_3Cu_{1.5}O_{5.5}$, and yttrium oxide (Figure S1).

### 3.3.2. Via Rietveld Refinement of 5:1 Tetramethylammonium Nitrate/YBaCu Nitrates

As a proof of concept, and due to the fact that the synthesis could not be done by a different synthetic pathway, namely, by avoiding the synthesis of metal carbonates using only ammonium nitrate and YBaCu nitrates in the reaction, tetramethylammonium nitrate was used. By the incorporation of carbon sources in the reaction, compared to the synthesis carried out by using ammonium nitrate/YBaCu nitrates, higher total mass percentage of the superconductive phase was expected to be formed.

The synthesis with tetramethylammonium nitrate goes as followed (Figure 5):

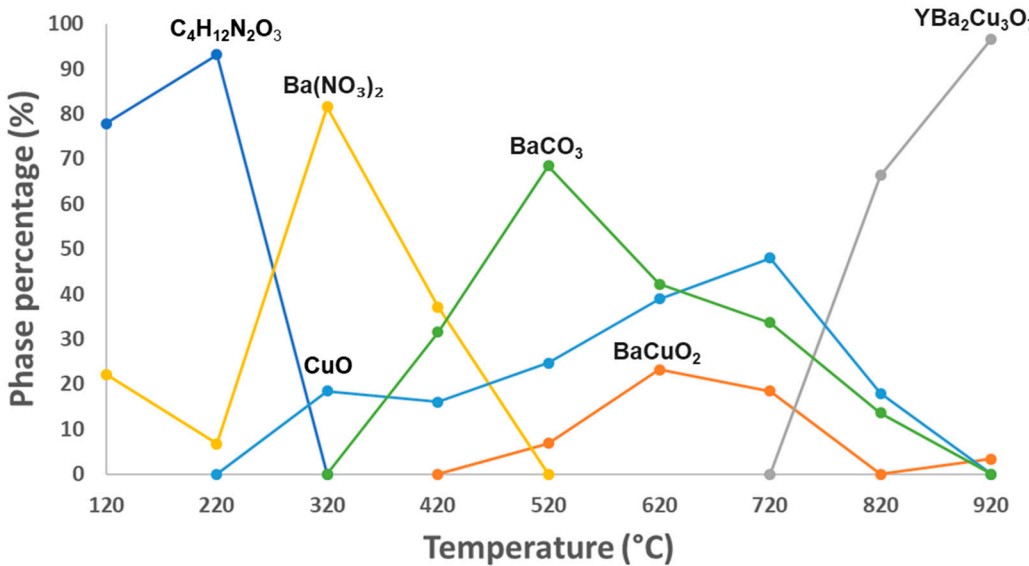

**Figure 5.** Graphical representation via qualitative analysis of pXRD patterns from the temperature analysis of the synthesis of YBCO (123) superconductor in the 5:1 molar ratio of tetramethyl ammonium nitrate/metal nitrates.

At the beginning of the temperature analysis, from 120 to 220 °C, tetramethyl ammonium nitrate was the majority phase with >80% of the total phase percentage across this temperature range. This phase is consumed on further heating with the majority of the phase percentage barium nitrate (81%), and a small percentage of copper oxide (19%). From 420 to 520 °C, barium nitrate continuously decomposes, from 37% to 0%, and hence, barium carbonate starts to form, present as 31% of the material at 420 °C and 68% at 520 °C. From that point onwards, barium carbonate will only decay in total phase percentage, and as a consequence, barium copper oxide will appear first at 520 °C. At this point, all the necessary precursor phases for the formation of YBCO are present in the system, namely, copper oxide, barium copper oxide, and barium carbonate. YBCO is first seen at 820 °C and subsequently is present as 96% at 920 °C. Powder diffraction patterns of the synthesis of the superconductive phase via 5:1 molar ratio of ammonium and tetramethyl ammonium nitrate can also be found in Figure S2. Additionally, tabulated data comparing the total mass percentage obtained by ammonium nitrate/YBaCu nitrates and tetramethylammonium nitrate/YBaCu nitrates can be found in Table S2.

### 3.4. Thermogravimetry Analysis (TGA)

Thermogravimetric analysis (TGA) (Figure 6a) and differential thermal analysis (DTA) (Figure 6b) of the reactions using 5:1 molar ratio eutectic mixtures of ammonium nitrate and tetramethyl ammonium nitrate in combination with YBaCu nitrates were also performed. For the analysis, neutral atmosphere in the form of argon gas was used. For a 5:1 molar ratio of ammonium nitrate to metal nitrate, there was a loss of 8% of total mass percentage, which was due to the early evaporation of ammonium nitrate and possible moisture adsorbed by the material. At 173 °C (Figure 6a I*), there was a significant loss in mass percentage from 93% to 40% due to the complete evaporation of ammonium nitrate. Consistently, in this period, DTA analysis showed exothermic behavior, viz. from 240 to 280 °C, highlighted in Figure 6b blue*. From 280 °C (Figure 6a II*) to 500 °C (Figure 6a III*) there was a gradual decay in mass percentage of only 5%. However, soon after there was a second sharp decay, from 35% at 500 °C to 18% at 622 °C, from the barium nitrate decomposition. In this range of temperatures, from 500 to 622 °C, DTA analysis showed endothermic behavior until the very end of the range of temperatures, from 610 to 622 °C, where exothermic behavior was again observed. From that point onwards, the TGA total mass percentage remained constant, as well as a continual endothermic behavior observed in DTA analysis.

Conversely, TGA on tetramethyl ammonium nitrate samples in combination with YBaCu nitrates, with the same 5:1 molar ratio, showed no differences on the mass percentage until 250 °C (Figure 6a I•) was reached. Shortly after, there was a considerable weight loss of 69%, from 99% to 30%, in the range of temperatures between 250 and 380 °C (Figure 6a II •). DTA analysis in this range of temperatures, from 250 to 380 °C, exhibited exothermic behavior from 270 to 375 °C, highlighted in Figure 6B *red, specifically represented in two peaks at 290 and 328 °C. Such weight loss and exothermic behavior can be attributed mainly to tetramethyl ammonium nitrate evaporating and nitrate from YBaCu nitrates as well. From that point onwards, there was a constant tendency to lose weight, dropping 7% more of total mass from 380 to 660 °C (Figure 6a III •), and an unvarying endothermic behavior observed by the DTA analysis. Subsequently, the decomposition of barium carbonate to release $CO_2$ afforded a further mass loss of 2%.

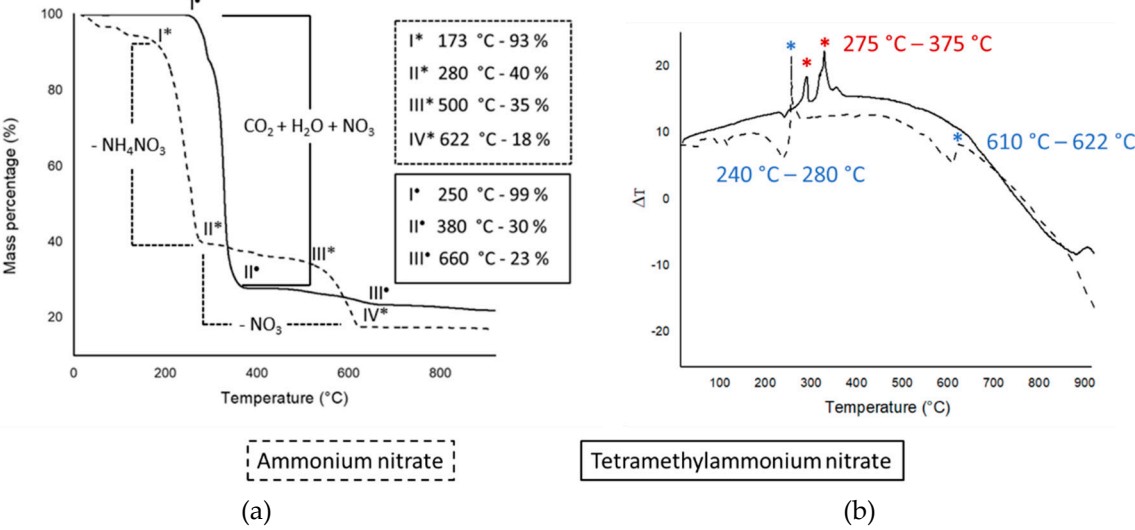

**Figure 6.** (**a**) TGA analysis, and (**b**) differential thermal analysis (DTA) analysis of the synthesis of YBCO (123) superconductor via 5:1 molar ratio of ammonium nitrate and tetramethyl ammonium nitrate/YBaCu nitrates.

## 4. Structural Characterization

Scanning electron microscopy was performed for samples collected when the reaction was carried out using tetramethylammonium nitrate/metal nitrates, in molar ratios varying from 20:1 to 1:20

(Figure S3), and at a maximum calcination temperature of 920 °C. From this, it was clear that there was a tendency for the size of crystallites to be reduced while the concentration of both compounds was in the range of 1:1≤5 (Figure 7b,c). Specifically, 1:1 was shown to be the most uniform, whereas 5:1, and 3:1 also exhibited similar sizes of crystallites, with the majority of the crystallites being around 0.5 to 1 μm, and with only slight amounts of bigger crystals with various sizes from above 1 to 7 μm. It was also noted that when the metal nitrates were in a higher molar ratio than tetramethyl ammonium nitrate, the crystallite sizes were increased considerably (Figure 7a).

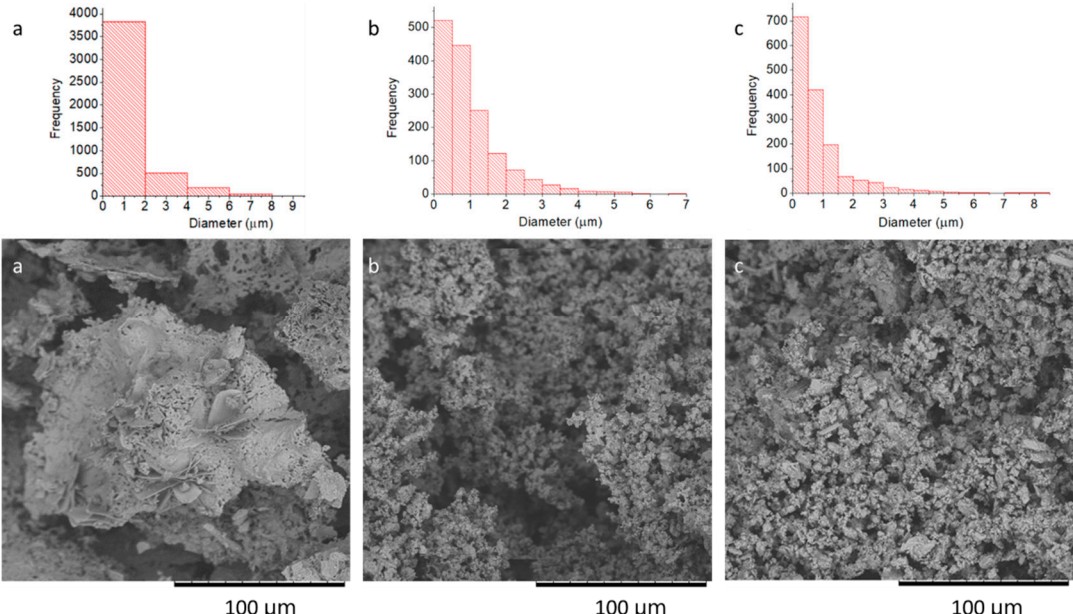

**Figure 7.** Histograms obtained via image analysis showing the particle size distribution obtained via (**a**) 5:1, (**b**) 1:1, and (**c**) 1:5 molar ratios of tetramethyl ammonium nitrate/YBaCu nitrates. (**a**–**c**) SEM images of their respective synthetic conditions. The reactions were carried out at a maximum temperature of 920 °C.

## 5. Conclusions

During this study, it has been proven that finding the eutectic point gives better crystalline results in terms of purity of the sample. This can be attributed to the fast mass transport that the system provides when a eutectic mixture is formed. Furthermore, highly reactive elements, such as barium, in the absence of carbonaceous material will react with the atmospheric $CO_2$, forming barium carbonate, and with Si, provided by the crucible, to form barium silicates, in order to form a stable configuration. Therefore, adding a carbon source to the reaction will improve the overall results by facilitating the formation of barium carbonate and restraining barium from reacting with undesired elements such as silicon contained in the crucible.

These results not only provide a better understanding of the use of ionic liquids/deep eutectic mixtures for the synthesis of metal oxides but have also provided greater clarity in the factors important to consider when searching for molecules to act as cations. Furthermore, the study here has supplied clarification on the relevance of metal carbonates during the syntheses of complex metal oxides produced via calcination in air.

**Supplementary Materials:** The following are available online at http://www.mdpi.com/2073-4352/10/5/414/s1.

**Author Contributions:** O.G.R. carried out experiments, data collection and writing of the article. T.N. and S.R.H. provided valuable guidance and important discussions which made the writing of this article possible. All authors have read and agreed to the published version of the manuscript.

**Funding:** This research received no external funding.

**Acknowledgments:** O.G. would like to thank Thi Mai Dung for valuable discussions and Suematsu Hisayuki for helping with TGA analysis. O.G. would also like to thank the WISE program for funding.

**Conflicts of Interest:** The authors declare no conflict of interest. The funders had no role in the design of the study; in the collection, analyses, or interpretation of data; in the writing of the manuscript, or in the decision to publish the results.

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
