# Peer review of "Synthesis of a Metal Oxide by Forming Solvate Eutectic Mixtures and Study of Their Synthetic Performance under Hyper- and Hypo-Eutectic Conditions"

_crystals, doi:10.3390/cryst10050414_

Round 1

Reviewer 1 Report

The submitted manuscript is a systematic study of the effect of different eutectic forming agents (and different hyper/hypo-eutectic conditions) on the synthesis of YBCO superconductor, aiming to identify possible cheaper and more effective way to produce it. The overall significance of the manuscript is quite high and I recommend to publish it on Crystals. However, the authors should deal with several issues and elaborate more on certain topics prior publishing.

In particular:

  1. please consider improving the introduction by citing more recent studies to strenghten ref. 1-7;
  2. since you evaluate the cost-effectiveness of different ammonium compounds, it should be interesting to add a brief comment on price differences among the analyzed compounds (I suppose ammonium carbonate is the cheapest);
  3. "... (150316) was purchased..." instead of was purchase;
  4. the synthesis procedure you followed is not quite clearly described in the way it actually is (from the "Generation of metal containing deep eutectic solvents" paragraph to the "heating protocols" one); please consider reformulating and detailing the whole synthesis procedure;
  5. it should be interesting to postulate within the text why it has been possible to obtain eutectic mixes with barium nitrate/ammonium nitrate;
  6. Figure 1: please use different colours or improve the differences in dashed lines' legenda;
  7. Figure 3: please substitute tetramethyl ammonium nitrate with "ammonium nitrate";
  8. Page 5, line 139: please use Figure 3 instead of Figure 2.
  9. I have to admit that it is surprising to observe such percentages of barium carbonate at 520 - 620 °C provided there's only the atmospheric CO2 as C source; have you tried (if it is possible for your instrumentation) to repeat the experiments in a controlled/inert atmosphere? Plus, where the Si come it from to form barium silicate? From the crucible?
  10. Page 7: "... ammonium nitrate was the majority..." instead of was thcorree;
  11. TGA analysis: the first 8% of weight loss could be related to residual incorporated water and/or moisture; additionally, you have to elaborate on the barium carbonate presence revealed by the XRD patterns but not detected by the TGA; finally, do you have DSC (or DTA) data coupled with your TGA data?

Author Response

  • Please consider improving the introduction by citing more recent studies to strenghten ref. 1-7;

Answer: References from 1 to 7 have been changed for more recent studies.

  • since you evaluate the cost-effectiveness of different ammonium compounds, it should be interesting to add a brief comment on price differences among the analyzed compounds (I suppose ammonium carbonate is the cheapest);

Answer: For comparison prices of all the products used were added as well as the price of an ionic liquid.

  • "... (150316) was purchased..." instead of was purchase;

Answer: Was purchase has been modified to was purchased.

  • the synthesis procedure you followed is not quite clearly described in the way it actually is (from the "Generation of metal containing deep eutectic solvents" paragraph to the "heating protocols" one); please consider reformulating and detailing the whole synthesis procedure;

Answer: To add more clarity the whole section of synthesis procedure has been rewritten.

  • it should be interesting to postulate within the text why it has been possible to obtain eutectic mixes with barium nitrate/ammonium nitrate;

Answer:

  • Figure 1: please use different colours or improve the differences in dashed lines' legenda;

Answer: The whole figure was reworked to add clarity. Mistakes on the legends were also corrected.

  • Figure 3: please substitute tetramethyl ammonium nitrate with "ammonium nitrate";

Answer: Figure 3 caption has been corrected accordingly from tetramethyl ammonium nitrate to ammonium nitrate.

  • Page 5, line 139: please use Figure 3 instead of Figure 2.

Answer: Figure 2 has been replaced with Figure 3, now is consistent with the text.

  • I have to admit that it is surprising to observe such percentages of barium carbonate at 520 - 620 °C provided there's only the atmospheric CO2 as C source; have you tried (if it is possible for your instrumentation) to repeat the experiments in a controlled/inert atmosphere? Plus, where the Si come it from to form barium silicate? From the crucible?

Answer: “barium silicate (B2SiO4), silicon being provided by the surface of the crucible.” Has been added in the text to add clarity.

A reaction in neutral atmosphere was performed initially but the text, likewise supplementary information suffered some changes when uploaded to the system. “To prove the formation of carbonates due to the interaction of barium with the carbon contained in the atmosphere a reaction under argon atmosphere was done. The outcome shown no carbonates in the least, although only weak noise/signal ratio diffractions are observed, namely yttrium barium copper oxide with the chemical composition Y0.5Ba3Cu1.5O5.5, and yttrium oxide (Supplementary information Figure S1).” has now been added to the text.

  • Page 7: "... ammonium nitrate was the majority..." instead of was thcorree;

Answer: The text has been corrected, now it reads “ammonium nitrate was the majority”

  • TGA analysis: the first 8% of weight loss could be related to residual incorporated water and/or moisture; additionally, you have to elaborate on the barium carbonate presence revealed by the XRD patterns but not detected by the TGA; finally, do you have DSC (or DTA) data coupled with your TGA data?

Answer: DTA analysis has been added. Also the text now reads “of 8% of total mass percentage, which is due to the early evaporation of ammonium nitrate and possible moisture adsorbed by the material.”

Reviewer 2 Report

The authors have investigated the synthesis of metal oxides through the eutectic formation. Overall, the results are interesting with great research analysis and data. In addition, the supporting documents were not provided to evaluate the manuscript. However, the author needs to address some of the content to better convey the results properly.

1) It is easy to get lost in the manuscript to which metal nitrate or YBC nitrate the author referring too. I request authors to carefully edit the manuscript to indicate the metal they are implying.

2) Line 66, replace element to metal nitrate, similarly in line 89

3) The eutectic point for copper nitrate is not consistent in the manuscript and figure 1

4) In figure 1, the lines for barium nitrate and the legends are not matching

5) Line 139, replace figure 2 to figure 3

6) There is no figure 3*, rather mention see an asterisk (*) indicated in figure 3

7) Figure 5, replace ammonium to tetramethyl ammonium

8) Figure 6, it should be barium nitrate than metal nitrate

9) From the TGA data, the difference between the barium carbonate obtained from ammonium nitrate/barium nitrate (18%) and tetramethylammonium nitrate/barium nitrate (21%). Assuming the excess mass is due to carbon presence, but the superconductive phase percentage is high. Could you briefly explain?

 10) Rewrite the conclusions

11) The supporting information document is missing.

Author Response

1) It is easy to get lost in the manuscript to which metal nitrate or YBC nitrate the author referring too. I request authors to carefully edit the manuscript to indicate the metal they are implying.

Answer: The article has been revised and corrected accordingly to add clarity to the text. Special care was placed in mentioning which metal nitrate is being used.

2) Line 66, replace element to metal nitrate, similarly in line 89

Answer: The word element has been replaced for metal nitrate as indicated.

3) The eutectic point for copper nitrate is not consistent in the manuscript and figure 1

Answer: The eutectic point for copper nitrate has been changed from 106 °C to 105 °C to be consistent with figure 1.

4) In figure 1, the lines for barium nitrate and the legends are not matching

Answer: Answer: The whole figure was reworked to add clarity. Mistakes on the legends were also corrected.

5) Line 139, replace figure 2 to figure 3

Answer: Figure 2 has been replaced with Figure 3, now is consistent with the text.

6) There is no figure 3*, rather mention see an asterisk (*) indicated in figure 3

Answer: figure 3* has been changed to “see an asterisk (*) indicated in figure 3”

7) Figure 5, replace ammonium to tetramethyl ammonium

Answer: Figure 5 caption has been corrected accordingly from ammonium to tetramethyl ammonium nitrate.

8) Figure 6, it should be barium nitrate than metal nitrate

Answer: Figure 6 shows the TGA/DTA analysis of ammonium nitrate mixed with all the metal nitrates. However, the figure was also reworked to add clarity. Also, caption has been changed from “metal nitrates” to “YBaCu nitrates.”

9) From the TGA data, the difference between the barium carbonate obtained from ammonium nitrate/barium nitrate (18%) and tetramethylammonium nitrate/barium nitrate (21%). Assuming the excess mass is due to carbon presence, but the superconductive phase percentage is high. Could you briefly explain?

Answer: TGA/DTA was performed under neutral atmosphere. The differences between percentages we assume are due to the formation and remaining of barium carbonate when tetramethyl ammonium nitrate is used, whereas the analysis being done using ammonium nitrate could not formed any barium carbonate.

10) Rewrite the conclusions

Answer: Conclusions section was rewritten highlighting the novelty and most relevant results of the article.

11) The supporting information document is missing.

Answer: Supporting information was added but it didn’t upload it correctly. This time we are ensuring that the supporting information will be available.

Reviewer 3 Report

The paper 'Synthesis of a metal oxide by forming solvate eutectic 2 mixtures and study of their synthetic performance 3 under hyper- and hypo-eutectic conditions' is interesting and well presented. However, the paper needs some revisions before it can be published. 

Some of the changes required are:

  1. Figure-1: These lines are confusing. I suggest authors use symbols and colors (at least online, colored lines) which would then help readers. 
  2. Minor English corrections are required: for instance, page-3, line-100: it should be 'to be formed'; page-4, line-119: was conducted were??; page-6, line-150: it should be were and not where!!
  3. Page-5, line-139: it should be Figure-3 and not Figure-2!
  4. Page-5, line-145: expand XMR!
  5. Page-8, line-208 and page-9, line-227: CO2, put 2 into subscript mode!
  6. I could not see the supplementary information. I link specified says nothing is present there!! Please check it and make sure it's live!

Author Response

  • Figure-1: These lines are confusing. I suggest authors use symbols and colors (at least online, colored lines) which would then help readers. 

Answer: Answer: The whole figure was reworked to add clarity. Mistakes on the legends were also corrected.

  • Minor English corrections are required: for instance, page-3, line-100: it should be 'to be formed'; page-4, line-119: was conducted were??; page-6, line-150: it should be were and not where!!

Answer:  The mistakes were corrected and where to be necessary the whole paragraph was rewritten.

  • Page-5, line-139: it should be Figure-3 and not Figure-2!

Answer: Figure 2 has been replaced with Figure 3, now is consistent with the text

  • Page-5, line-145: expand XMR!

Answer: XMR was deleted and corrected accordingly, now it reads “at 420 °C with only 8 % but with a positive tendency on the next 200 °C, covering 55 % and 59 % of the total mass at 520 °C and 620 °C respectively. “

  • Page-8, line-208 and page-9, line-227: CO2, put 2 into subscript mode!

Answer: Both CO2 have now the correct formulation (CO2)

  • I could not see the supplementary information. I link specified says nothing is present there!! Please check it and make sure it's live!

Answer: Supporting information was added but it didn’t upload it correctly. This time we are ensuring that the supporting information will be available.

Round 2

Reviewer 2 Report

Line 73-75 “However, from all the ammonium compounds used eutectic mixtures were only possible to be produced using ammonium nitrate and  tetramethyl ammonium nitrate.”

After mixing ammonium nitrate and  metal nitrate, do they instantly formed eutectic mixture, or any external source like the heat is required? Detail procedure for experiments was missing?

Aqueous mixtures are directly mixed with the ammonium compounds?

Line 117-119: The mix of metal nitrates combined with ammonium nitrate derived in a eutectic mixture at 100 °C in a molar ratio of 5:1 ammonium nitrate/YBaCu nitrates, six-degree Celsius below copper nitrate. Despite barium nitrate not forming a eutectic solvent at that molar ratio, no precipitates were observed

Does the author mean YBC for a mix of metal nitrates? Five degree Celsius, not six degrees

I would re-write the sentence as “ Combination of YBaCu nitrate mixture and ammonium nitrate at 5:1 molar ratio resulted deep melting point at 100 C, which is five degrees lower than the Copper nitrate and ammonium nitrate eutectic mixture.

Source/s for the prices of each ammonium compounds?

Line 93-95 “Despite not all the mixes of ammonium compounds with metal nitrates exhibited eutectic behaviour, all the mixes were calcined. All the mixes of metal nitrates and ammonium compounds were firstly dehydrated for 24 h before calcining. After dehydration the calcining processes followed the same calcination route: max temperature 920 °C at a ramp rate 5 °C/min with a dwelling time of 2 h.

Please correct the sentences they are grammatically incorrect and do it for the rest of the document

Figure S2 and Table S2; replace metal nitrates to YBCu nitrates

Line 225-227; Thermogravimetric analysis (TGA) (Figure 6A) and differential thermal analysis (DTA) (Figure 6B) of the reaction using ammonium nitrate and tetramethyl ammonium nitrate in combination with the metal nitrates was also performed.

These are eutectic mixtures. I would re-phrase the sentence.

What is the difference between Figure S1 and Figure S2A?

The font sizes in the tables of supporting information are small.

Figure 7, Specific the temperature at which YBCu nitrate and ammonium nitrate eutectic performed in the figure caption and the respective section?

The phase percentage of superconductor mixtures increase from 620 to 920 degrees Celsius. The author has shown the crystalline nature and morphology of the mixture. However, some of the details of the experiment were missing like temperature.

I am curious to see the crystal and morphology changes of these superconductor mixtures from 620 to 920 using XRD and SEM images. Are there any drastic changes in the morphology of 5:1 mixture at these four temperatures 620, 720, 820, and 920 degrees Celsius?

Specific details of characterization techniques were missing. For instance, what pan was used to measure the thermal stability of these mixtures

Author Response

  • Line 73-75 “However, from all the ammonium compounds used eutectic mixtures were only possible to be produced using ammonium nitrate and  tetramethyl ammonium nitrate.”

After mixing ammonium nitrate and metal nitrate, do they instantly formed eutectic mixture, or any external source like the heat is required? Detail procedure for experiments was missing?

Aqueous mixtures are directly mixed with the ammonium compounds?

Answer: The segment has been further described, now it reads “Initially, the aqueous solutions are mixed correspondingly, and left at 70 °C for a period of 24 hrs to ensure full dehydration. After that, solid products are obtained in all the mixes of ammonium compounds with every single metal nitrate, in all the range of molar ratios. The eutectic point is then noted by raising the temperature until the solid becomes entirely liquid.”

Also, the generation of aqueous precursors has been modified to “Yttrium nitrate (0.05 M), barium nitrate (0.1 M), and copper nitrate (0.15 M) were individually mixed in vials with DI water under stirring until all salts are dissolved. Similarly, ammonium compounds, i.e. ammonium nitrate (0.5 M), formate, acetate (0.5 M), carbonate (0.5 M), bicarbonate (0.5 M), and tetramethylammonium nitrate (0.5 M), and formate (0.5 M), are also dissolved under constant stirring with DI water until all the solid material is fully dissolved.”

  • Line 117-119: The mix of metal nitrates combined with ammonium nitrate derived in a eutectic mixture at 100 °C in a molar ratio of 5:1 ammonium nitrate/YBaCu nitrates, six-degree Celsius below copper nitrate. Despite barium nitrate not forming a eutectic solvent at that molar ratio, no precipitates were observed

Does the author mean YBC for a mix of metal nitrates? Five degree Celsius, not six degrees

I would re-write the sentence as “ Combination of YBaCu nitrate mixture and ammonium nitrate at 5:1 molar ratio resulted deep melting point at 100 C, which is five degrees lower than the Copper nitrate and ammonium nitrate eutectic mixture.

Answer: Thank you for the suggestion, the paragraph now reads “Combination of YBaCu nitrate mixture and ammonium nitrate at 5:1 molar ratio resulted deep melting point at 100 C, which is five degrees lower than the Copper nitrate and ammonium nitrate eutectic mixture.”

  • Source/s for the prices of each ammonium compounds?

Answer: It was added to the text “All the prices shown are from Sigma Aldrich JP”, the paragraph now reads “(Tetramethylammonium nitrate ¥29,600/100g, Tetramethylammonium formate ¥37,357/100g, ammonium nitrate ¥1,664/100g, ammonium acetate ¥1,828/100g, ammonium carbonate ¥6,120/100g, ammonium bicarbonate ¥3,640/100g, ammonium formate ¥12,200/100g. All the prices shown are from Sigma Aldrich JP)”

  • Line 93-95 “Despite not all the mixes of ammonium compounds with metal nitrates exhibited eutectic behaviour, all the mixes were calcined. All the mixes of metal nitrates and ammonium compounds were firstly dehydrated for 24 h before calcining. After dehydration the calcining processes followed the same calcination route: max temperature 920 °C at a ramp rate 5 °C/min with a dwelling time of 2 h.

Please correct the sentences they are grammatically incorrect and do it for the rest of the document

Answer: The paragraph has been changed to “All the mixes of ammonium compounds with metal nitrates were calcined following the same calcination route: max temperature 920 °C at a ramp rate 5 °C/min with a dwelling time of 2 h.”

  • Figure S2 and Table S2; replace metal nitrates to YBCu nitrates

Line 225-227; Thermogravimetric analysis (TGA) (Figure 6A) and differential thermal analysis (DTA) (Figure 6B) of the reaction using ammonium nitrate and tetramethyl ammonium nitrate in combination with the metal nitrates was also performed.

Answer:  Where the metal nitrates implied the use of the mix of yttrium nitrate, barium nitrate, and copper nitrate, has been changed to to YBaCu nitrates.

  • These are eutectic mixtures. I would re-phrase the sentence.

Answer:  The paragraph now reads “Thermogravimetric analysis (TGA) (Figure 6A) and Differential thermal analysis (DTA) (Figure 6B) of the reactions using 5:1 molar ratio eutectic mixtures of ammonium nitrate and tetramethyl ammonium nitrate in combination with YBaCu nitrates were also performed.”

  • What is the difference between Figure S1 and Figure S2A?

Answer: Figure S1 shows the powder diffraction pattern of the synthesis of YBCO (123) superconductor via 5:1 molar ratio of ammonium nitrate/YBaCu nitrates carried out under neutral atmosphere (Ar) at a max temperature of 620 °C.

Figure S2 shows the powder diffraction patterns of the synthesis of YBCO (123) superconductor via 5:1 molar ratios of A) ammonium nitrate/ YBaCu nitrates, and B) tetramethyl ammonium nitrate/ YBaCu nitrates carried out in air atmosphere and at max temperature of 920 °C.

  • The font sizes in the tables of supporting information are small.

Answer: The font size follows the indicated by the template provided by the magazine (Times new roman 9)

  • Figure 7, Specific the temperature at which YBCu nitrate and ammonium nitrate eutectic performed in the figure caption and the respective section?

Answer: The sentence “The reactions were carried out at a max temperature of 920 °C.” has been added.  

  • The phase percentage of superconductor mixtures increase from 620 to 920 degrees Celsius. The author has shown the crystalline nature and morphology of the mixture. However, some of the details of the experiment were missing like temperature.

Answer: The calcination temperature has been added, now it reads “Scanning electron microscopy was performed for samples collected when the reaction was carried out using tetramethylammonium nitrate/metal nitrates, in molar ratios varying from 20:1 to 1:20 (Supplementary Information Figure S3), and at a max calcination temperature of 920 °C”

  • I am curious to see the crystal and morphology changes of these superconductor mixtures from 620 to 920 using XRD and SEM images. Are there any drastic changes in the morphology of 5:1 mixture at these four temperatures 620, 720, 820, and 920 degrees Celsius?

Answer: Unfortunately, we cannot provide this information yet but, it is something we are further analyzing.

  • Specific details of characterization techniques were missing. For instance, what pan was used to measure the thermal stability of these mixtures

Answer: The hot plate has now been specified in the Generation of metal containing deep eutectic solvents. Now it reads “The eutectic point is then noted by raising the temperature, using a fisher scientific isotemp advance hot plate stirrer, until the solid becomes entirely liquid.”